# Self-filtering narrowband high performance organic photodetectors enabled by manipulating localized Frenkel exciton dissociation

Boming Xie[1,2], Ruihao Xie[1,2], Kai Zhang [1✉], Qingwu Yin[1], Zhicheng Hu[1], Gang Yu[1], Fei Huang [1✉] & Yong Cao[1]

The high binding energy and low diffusion length of photogenerated Frenkel excitons have long been viewed as major drawbacks of organic semiconductors. Therefore, bulk hetero-junction structure has been widely adopted to assist exciton dissociation in organic photon-electron conversion devices. Here, we demonstrate that these intrinsically "poor" properties of Frenkel excitons, in fact, offer great opportunities to achieve self-filtering narrowband organic photodetectors with the help of a hierarchical device structure to intentionally manipulate the dissociation of Frenkel excitons. With this strategy, filter-free narrowband organic photodetector centered at 860 nm with full-width-at-half-maximum of around 50 nm, peak external quantum efficiency around 65% and peak specific detectivity over $10^{13}$ Jones are obtained, which is one the best performed no-gain type narrowband organic photodetectors ever reported and comparable to commercialized silicon photodetectors. This novel device structure along with its design concept may help create low cost and reliable narrowband organic photodetectors for practical applications.

[1] State Key Laboratory of Luminescent Materials and Devices, Institute of Polymer Optoelectronic Materials and Devices, School of Materials Science and Engineering, South China University of Technology, 381 Wushan Road, 510640 Guangzhou, China. [2] These authors contributed equally: Boming Xie, Ruihao Xie. ✉email: mszhangk@scut.edu.cn; msfhuang@scut.edu.cn

Narrowband photodetectors, which only detect light within a specific wavelength of interest and have no response to light of other wavelengths (usually background or environmental radiation), have been widely used in imaging, chemical analysis, and have also shown great potential for application in emerging artificial intelligence networks, such as augmented/virtual reality, advanced driver assistance systems and full-weather robots[1–3]. Commercially available narrowband photodetectors are made of inorganic semiconductors (mostly silicon) and integrated with optical filters[4]. The use of filter creates additional optical interfaces, increases the device's architectural complexity and limits the pixel density of the detector array[5]. Moreover, due to the small extinction coefficient of silicon, a thick silicon film (~3 μm) is often required to sufficiently absorb light for operation, which results in an array with a greater likelihood of interpixel cross-talk. To overcome these defects, several strategies are emerging for narrowband photodetection without filter, including: (1) using absorbers with narrowband absorption[6,7]; (2) intentionally enhancing the absorption in a selected wavelength range by the plasmonic effect[8–10]; and (3) manipulating the internal quantum efficiency via charge collection narrowing (CCN)[11–13]. These pioneer works provided effective strategies of making filter-free narrowband photodetectors. Nevertheless, in order to suppress the response of the background light and realize the narrowband characteristic, most of these methods have to sacrifice the original sensitivity of the photodetectors and the resulting devices' performances are still behind those of silicon photodetectors.

Among all kinds of semiconductors, organic semiconductors have high extinction coefficients and can efficiently absorb light at thickness as little as a few hundred nanometers. The absorption spectrum of organic semiconductors can be easily tuned by adjusting their molecular structures[14]. In addition, because organic semiconductors are light-weight, flexible, and soluble in organic solvents, they are more suitable for wide applications that require inexpensive, flexible, and moldable photodetectors[15–18]. However, the typical high binding energy and short diffusion length of the excitons in organic semiconductors bring some challenges for their applications in photon–electron conversion devices. Unlike its inorganic counterparts, when an organic semiconductor absorbs a photon, a localized Frenkel exciton is created due to the weak intermolecular van der Waals interaction alongside with the low dielectric constant[19,20]. This exciton cannot spontaneously separate into free charge due to the strong Coulombic interaction[21], and the short lifespan of exciton limits its diffusion length within tens of nanometers[22]. Therefore, many strategies, including molecular structure modification to improve dielectric constant[23–25], and matching materials with different electron affinities (ionization potentials) to provide donor–acceptor (D/A) interfaces[26,27], have been adopted to assist exciton dissociation before dissipation in organic photon–electron conversion devices.

Herein, we find that these intrinsic "poor" properties of Frenkel excitons actually offer organic semiconductors great opportunities to achieve filter-free narrowband organic photodetectors (OPDs). We propose a simple strategy to produce narrowband OPDs by manipulating exciton dissociation (named as "exciton dissociation narrowing" [EDN]), with a hierarchical device structure where thick larger bandgap donor layers followed by a lower bandgap acceptor layer. During the operation, excitons generated by high-energy photons in donor front layers fail to separate into free charges due to the absence of D/A interfaces, thus dissipated. Only low-energy photons with a long penetration depth can reach the D/A interfaces and produce free charges for collection. Some other strategies based on thick active layer with bulk heterojunction (BHJ) structure have also been proposed to construct narrowband photodetectors, including no-gain type[11] and gain type[28,29]. For example, the no-gain type narrowband detectors enabled by CCN concept modulate the external quantum efficiency (EQE) spectrum by manipulating the collection efficiency of free charges at the corresponding electrodes (photogenerated excitons firstly dissociate into free charges). The gain type detectors achieve photomultiplication via charge tunneling injection from external circuit under large applied voltage. In comparison, the novel device structure and its generic methodology proposed in this study is based on hierarchical active-layer structure and novel working mechanism of EDN. It was interestingly found that this novel methodology can efficiently suppress the response outside the detection window while retaining high sensitivity in the detection region, which enabled us to produce a series of simple structure visible-blind near-infrared (NIR) narrowband OPDs with outstanding performances.

## Results

**Basic device structure of self-filtering narrowband OPD.** Figure 1a shows the chemical structures of the materials[30–33] used in this study. NT812 is a home designed high charge mobility (~$10^{-2}$ cm$^2$ V$^{-1}$ s$^{-1}$) donor polymer which shows ideal charge transport in organic solar cells even when the thickness reach to 1 μm[30,34]. The most crucial part of the proposed self-filtering (SF) narrowband OPD is the hierarchical structure of a thick larger bandgap donor layer followed by a lower bandgap acceptor layer. This hierarchical structure is usually difficult to be realized because the subsequent acceptor material will penetrate into the underlying donor layer network during solution processing[35,36]. To this end, a combination of methods was adopted. Firstly, to prevent the diffusion of the acceptor molecules (Y6, Fig. 1a), the donor polymer NT812 was crosslinked to be a robust film (with a thickness of 650 nm) by the azide crosslinker s-4PFA, which can crosslink semiconducting polymers with a sufficiently low concentration and has negligible influence on their crucial semiconductor properties[37]. Following that, a thin film of NT812 (100 nm) was deposited onto crosslinked NT812 to provide an interdiffusion region with following Y6 for efficient charge separation[38]. Then Y6 was deposited onto NT812 as the rear-layer to obtain a complete activelayer with a total thickness of 800 nm. Here, chloroform was selected as the solvent for rear-layer processing, in which Y6 exhibits good solubility, whereas the front-layer NT812 is nearly insoluble (as shown in Supplementary Fig. 1). This orthogonal solvent and its low boiling point further help the construction of the hierarchical structure.

To verify the existence of this hierarchical structure, we carried out the time-of-flight secondary ion mass spectrometry (ToF-SIMS) depth profiling measurement[39]. The C$_{74}$H$_{69}$F$_4$N$_8$O$_2$S$_5^-$ (fragment structure is shown in Supplementary Fig. 2) and the SN$^-$ ions were selected to track the depth distribution of Y6 and NT812, respectively. Figure 1b shows the ToF-SIMS depth profile through the prepared film. As expected, Y6 signal shows strong intensity at the first 130 nm, and begins to fall after 130 nm and tends to be zero after 168 nm. Meanwhile, the intensity of SN$^-$ ion begins to rise after 130 nm and stabilize after 168 nm, which we speculate it is due to the higher thiadiazole content within NT812 leading to higher SN$^-$ ion yield. Figure 1c shows the reconstructed three-dimensional chemical images of the C$_{74}$H$_{69}$F$_4$N$_8$O$_2$S$_5^-$ and SN$^-$ ions, which exhibit evidently that Y6 molecules are concentrated in the upper layer of the whole film with a penetration depth of 168 nm. These results demonstrate that using the methods described above, we effectively restricted the diffusion of Y6 molecules into the underlying donor layer, guarantee a complete device structure of ITO/PEDOT:PSS/donor front layer NT812/acceptor rear layer Y6/PFN-Br/Ag (as illustrated in Fig. 1d).

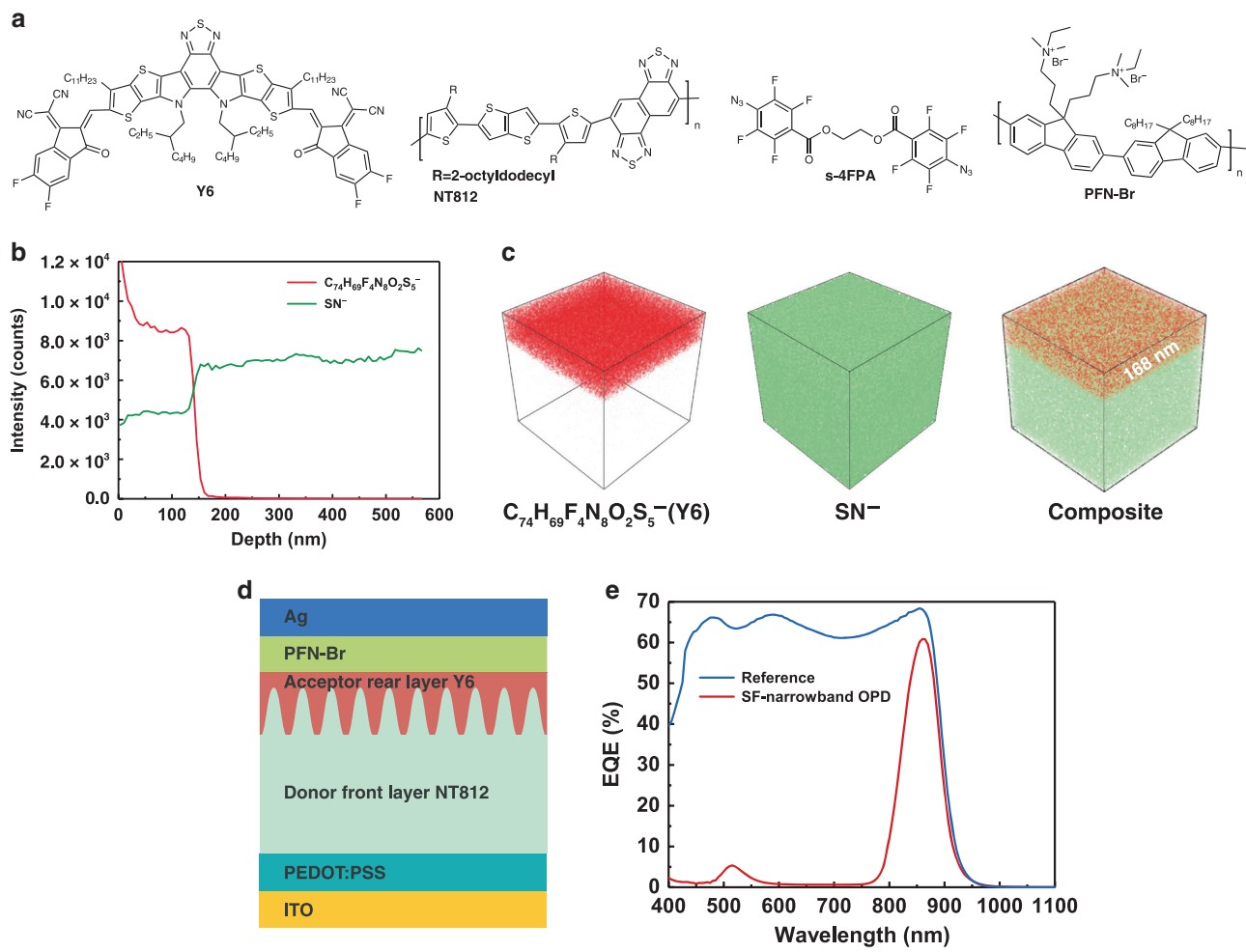

**Fig. 1 Hierarchical structure and EQE response of OPDs. a** Chemical structures of Y6, NT812, s-4FPA and PFN-Br. **b** Time-of-flight secondary ion mass spectrometry (ToF-SIMS) depth profile through the prepared film. **c** Reconstructed three-dimensional negative ion images (100 μm × 100 μm area) of $C_{74}H_{69}F_4N_8O_2S_5^-$, $SN^-$ and their composite from a depth profile of the prepared film. **d** Basic device structure of self-filtering (SF) narrowband organic photodetector (OPD). **e** External quantum efficiency (EQE) curves of the SF-narrowband OPD and reference device under −0.1 V bias.

Figure 1e presents the EQE spectrum of the fabricated photodetector under −0.1 V bias, which exhibits a dominant narrow peak at 860 nm accompanied by a weak response at 520 nm. Figure 1e also gives the EQE spectrum of the reference device with a conventional structure (ITO/PEDOT:PSS/NT812: Y6 (150 nm)/PFN-Br/Ag). EQE curves of the SF-narrowband OPD and the reference device show that, as the corresponding EQE response in visible spectrum is effectively suppressed in SF-narrowband OPD, the peak EQE at 860 nm is not significantly reduced, maintaining more than 89% of the corresponding value of reference device. By comparison, the BHJ device[11] with device struture of ITO/PEDOT:PSS/NT812:Y6 (800 nm, 1:4)/PFN-Br/ Ag suffers tremendous EQE loss, only obtains a peak EQE around 10% (as demonstrated in Supplementary Fig. 3).

**Working mechanism of self-filtering narrowband OPDs.** To understand the mechanism of EDN, the distribution of photo-generated excitons in the bulk of the front-layer material NT812 was analyzed. The continuity equation for neutral excitons can be described by Eq. (1):

$$\frac{\partial n(x,t)}{\partial t} = g\alpha N_0(t)e^{-\alpha x} - \frac{n(x,t)}{\tau} + D\frac{\partial^2 n(x,t)}{\partial x^2} - F(x - x_{int})n(x,t), \quad (1)$$

where $n(x,t)$ is the time-dependent exciton density, $x$ is the penetration depth of the incident light from the transparent

electrode in the bulk of the front layer, $g$ is the internal efficiency of photon-to-exciton, $N_0(t)$ is the number of incident photons, $\alpha$ is the absorption coefficient of the front layer material, $\tau$ is the exciton lifetime, $D$ is the exciton diffusion coefficient, and $F(x - x_{int})$ is the exciton dissociation rate at the D/A interface ($x_{int}$). Equation (1) is solved with the stationary illumination ($(\partial n/\partial t) = 0$) and boundary conditions prosed by Stübinger et al.[40]:

$$n(x) = \frac{gN_0}{D}\frac{\alpha L^2}{1 - (\alpha L)^2}\left(e^{-\alpha x} - e^{-(x/L)}\right), \quad (2)$$

where $L$ is the one-dimensional diffusion length, which is defined as $L = \sqrt{D\tau}$. Equation (2) offers the exciton density distribution in the bulk of the donor front layer. Furthermore, the exciton diffusion coefficient $D$ of the front-layer material NT812 is calculated by using the Monte Carlo simulation method[41]. (The specific simulation details are recorded in Supplementary Note 1). Figure 2a shows the experimentally measured time-resolved photoluminescence of the pristine and blend film with a $PC_{61}BM$ volume fraction of 0.05% for NT812 (open circles), and the photoluminescence decay of blend film, which was modeled with the Monte Carlo simulation, is also depicted as solid lines. The model fits the experimental data very well and yields the initial diffusion coefficient $D$ of $19.5 \times 10^4$ cm$^2$ s$^{-1}$; the diffusion length

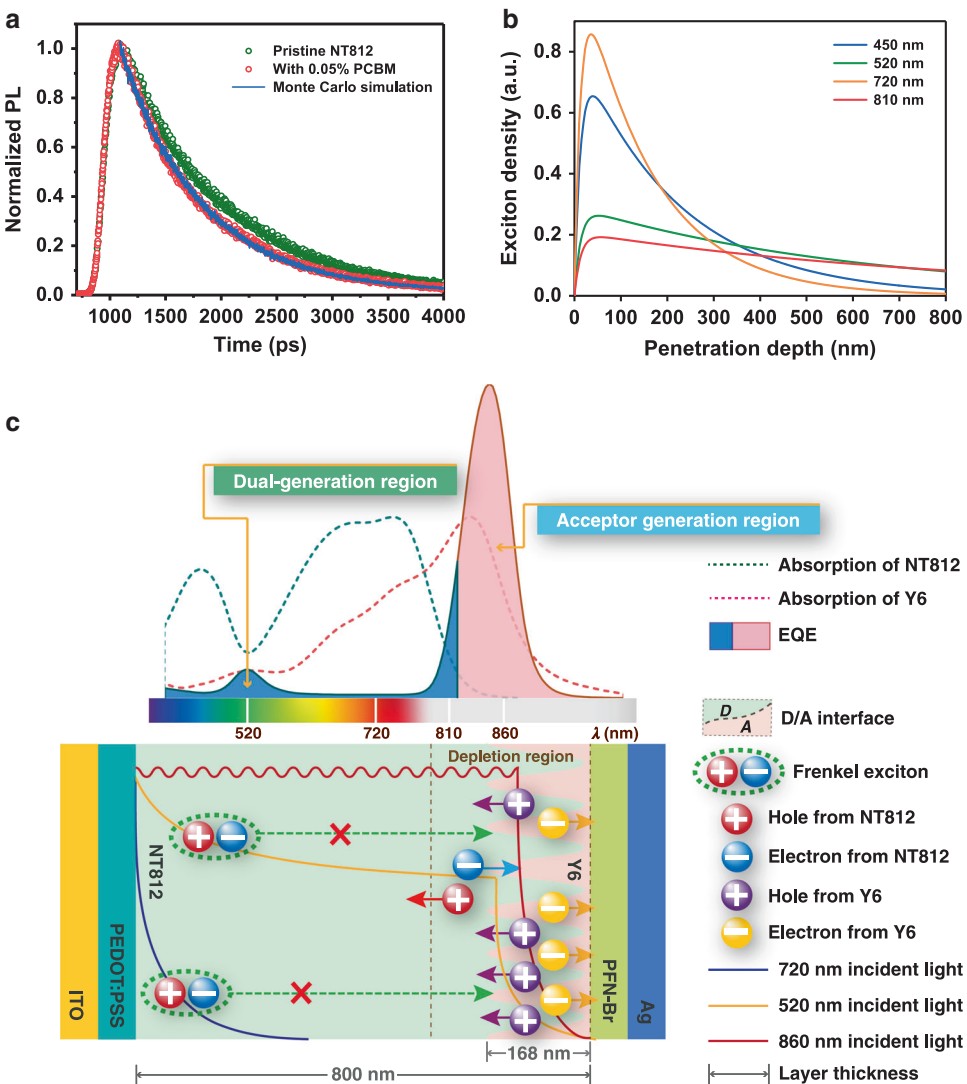

**Fig. 2 Working mechanism of SF-narrowband photodetectors. a** Experimentally measured time-resolved photoluminescence of pristine (green open circles) and blend film (red open circles). Photoluminescence decay modeled with the Monte Carlo simulation of blend film (blue solid lines). **b** Exciton density in the bulk of NT812 versus penetration depth of incident light with different wavelength from the transparent ITO electrode. **c** Diagrammatic illustration of the working mechanism of self-filtering (SF) narrowband photodetectors.

$L$ was then calculated as 13.3 nm. Combined with the value of $\alpha$ measured from the ultraviolet–visible (UV–vis) light absorption spectrum (Supplementary Fig. 4), the exciton density distribution curves of incident light of various wavelengths can be eventually obtained (Fig. 2b). From the distribution of excitons in the donor front layer, the working mechanism of the SF-narrowband OPD can be summarized as follows (as illustrated in Fig. 2c).

For incident light within the absorption range of the front donor material (i.e., high-energy photons in the spectral range of 380–850 nm for NT812), the corresponding photogenerated excitons are concentrated mainly on the front side of the film (e.g., within the first 200 nm layer, as shown in Fig. 2b), where the distance between these excitons and the D/A interfaces is far beyond the exciton diffusion length. During their limited lifetime, most excitons outside the depletion region cannot diffuse to the D/A interface to dissociate into free charges; therefore, the excitons generated by this spectral range of incident photons are bound in the bulk of the front donor material and eventually relax back to the ground state[42,43]. Moreover, according to the Beer–Lambert law[44], the intensity of incident light decreases exponentially with the penetration depth in the front layer. Thus,

most of the high-energy incident photons were absorbed by the thick front layer and cannot reach the acceptor rear layer (as shown by transmittance spectrum in Supplementary Fig. 5). As a result, neither the donor nor the acceptor material can contribute to the EQE in this spectral range of incident light. (as is typically represented by 720 nm incident light in Fig. 2b, c).

For incident light with a longer wavelength (longer than the absorption onset of the front donor material; i.e., $\lambda > 850$ nm for NT812), the corresponding low-energy photons can penetrate the entire front layer and be harvested by the rear acceptor material with tiny loss. The excitons generated in the acceptor material then efficiently dissociate and separate into free charges at the D/A interfaces and are collected by the corresponding electrodes, so the corresponding EQE response is obtained at the long wavelength spectrum, which we call the acceptor generation region, which is the main contribution to the narrow response peak. (As that of 860 nm incident light in Fig. 2c).

However, organic optoelectronic materials generally do not exhibit uniform absorption coefficients over a wide spectral range, photons within a specific spectral region (e.g., 520 and 810 nm for NT812; see Supplementary Fig. 4) can still penetrate deeply into the

bulk of the donor front layer and create a small number of excitons within the depletion region (as shown in Fig. 2b). This portion of the incident photons can also partially reach the acceptor rear layer and create excitons for dissociation due to the low absorption coefficient of the front donor material (see Supplementary Fig. 5). Thus, the weak EQE response that corresponds to these incident spectral ranges can be attributed to both the donor and acceptor layer, which we call the dual-generation region.

**Self-filtering narrowband OPD with double donor layers.** To further suppress the EQE response within the dual-generation region (spectra range of 500–600 nm), poly(3-hexylthiophene) (P3HT)[45], which provides a cascade highest occupied molecular orbital (HOMO) level with NT812 (see Fig. 3a), was selected to replace PEDOT:PSS as the SF hole transport layer (SF-HTL). P3HT was also crosslinked by s-4FPA to prevent it from being washed away by the following solution, as verified by the cross-section scanning electron microscope image (Fig. 3b). The double donor layers that containing NT812 and P3HT delivered a complete device structure of ITO/SF-HTL P3HT/donor front layer NT812/acceptor rear layer Y6/PFN-Br/Ag.

As shown in the UV–vis light absorption spectra (Fig. 3c), the absorption peak of P3HT is around 520 nm, which just covers the dip area of the NT812 absorption spectrum. As a result, the EQE response between 500 and 600 nm was successfully wiped out, while the narrowband EQE peak at 860 nm was not affected (as shown in Fig. 3d). Here, we define two parameters, out-of-band suppression factor $S = \frac{nor.R_{ref}(\lambda)}{nor.R_{SF}(\lambda)}$ and in-band transmission factor $T = \frac{R_{SF}(\lambda_0)}{R_{ref}(\lambda_0)}$, to characterize the self-filtering property of the narrowband photodetector, where $nor.R_{ref}(\lambda)$ and $nor.R_{SF}(\lambda)$ are the normalized responsivity of the reference device and SF-narrowband OPD, respectively. $R_{ref}(\lambda_0)$ and $R_{SF}(\lambda_0)$ are the responsivity of the reference device and SF-narrowband OPD at the narrow respond peak, respectively. Results are shown in Supplementary Fig. 6. The maximum out-of-band suppression factor at visible region is over 170, while the in-band transmission factor at 860 nm is 89.2%. Moreover, the spectral selectivity of our SF-narrowband OPD is based on the manipulation of excitons rather than free charge carriers after exciton dissociation[11]. Due to the high binding energy and the electrical neutrality of Frenkel exciton, the effect of applied electric field on the exciton diffusion is negligible[46]. Consequently, the EQE spectra of SF-narrowband

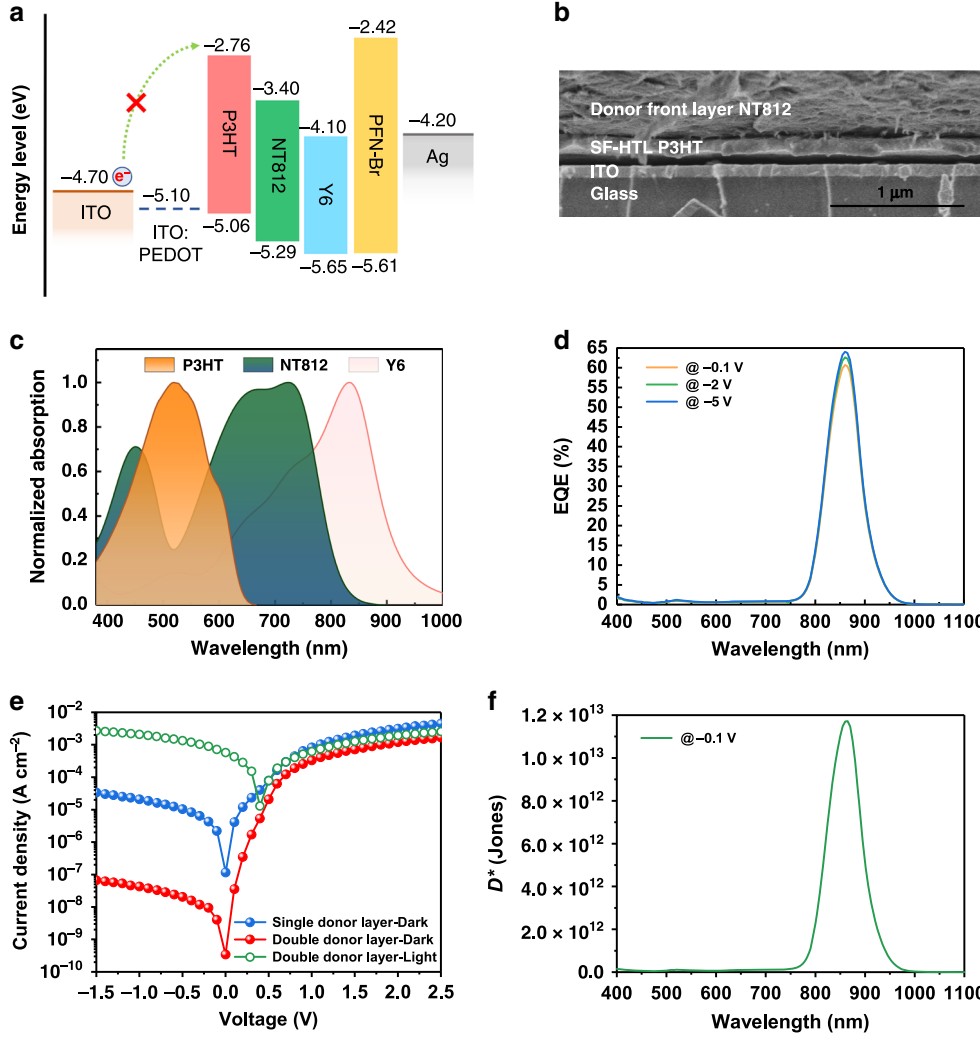

**Fig. 3 Device performances of SF-narrowband OPD with double donor layers. a** Energy level diagram of materials used to construct self-filtering (SF) narrowband organic photodetector (OPD) with double donor layers. **b** Cross-section scanning electron microscope image of the double donor layer structure, the scale bar identifies a length of 1 μm. **c** Normalized ultraviolet-visible (UV–Vis) absorption spectra of P3HT, NT812, and Y6. **d** External quantum efficiency (EQE) curves (at 165 Hz) of the SF-narrowband OPD with double donor layers under different voltage biases as indicated. **e** Current density–voltage (J–V) curves in dark and under illumination. **f** Specific detectivity spectra obtained from dark current density ($J_d$).

OPDs tends to saturate at higher applied voltages and retain spectral selectivity even under −10 V bias (Supplementary Fig. 7), which is so significant to maintain the consistency of narrowband characteristics in practical applications.

In addition to the narrow spectral response under illumination, the dark current density ($J_d$) is an important feature that has a pronounced impact on specific detectivity ($D^*$). As a fundamental source of electronic noise, the dark current in organic semiconductor materials is derived mainly from charge injection from the metal contacts under an applied external bias[47]. This injection phenomenon becomes more serious when the bandgap of the semiconductor material becomes smaller due to the reduced gaps (hence the reduced injection barrier) between material energy levels and the electrode Fermi levels, which is one of the main factors that limits the application of narrow-bandgap organic semiconductor materials in NIR photodetectors. Figure 3e shows the current density–voltage ($J$–$V$) characteristics of the two OPDs. Compared with the single donor layer device, the $J_d$ of the double donor layer detector is nearly three orders of magnitude lower under reverse bias, which should be ascribed to the excellent electron blocking ability[48] of wide-bandgap P3HT (Fig. 3a). Meanwhile, the cascade HOMO levels of the donor layers are favorable for the transport of photogenerated holes[49], and the hole mobility of SF-HTL P3HT (150 nm)/donor front layer NT812 (750 nm) was $1.02 \times 10^{-3}$ cm$^2$ V$^{-1}$ s$^{-1}$ as demonstrated in Supplementary Note 2 and Supplementary Fig. 8, so the desired EQE peak can be maintained[50,51]. With the incorporation of a SF-HTL P3HT, the obtained photodetectors demonstrated a peak specific detectivity $D^*$ of $1.2 \times 10^{13}$ Jones at 860 nm under −0.1 V bias (as shown in Fig. 3f) calculated from the expression of $D^* = R/\sqrt{2qJ_d}$, where $R$ is the responsivity (see Supplementary Fig. 9), and $q$ is the elementary charge of the electron (When the thermal noise is taken into consideration, the calculated $D^*$ slightly decreases to $9.5 \times 10^{12}$ Jones at 860 nm, see Supplementary Note 3). The specific detectivity curve on a logarithmic scale in Supplementary Fig. 10 shows that the specific detectivity at the detection peak is two orders of magnitude higher than that outside the detection window. The above calculation method assumes that the total noise of the photodetector is dominated by the shot noise in $J_d$. When considered other noise, such as thermal and $1/f$ noise, and measured the actual noise spectrum directly, a lower detectivity could be obtained according to the equation $D^* = R \times \sqrt{A}/S_n$, where $A$ is the device area and $S_n$ is the noise spectral density, yielding a peak specific detectivity $D^*$ of $2.4 \times 10^{12}$ Jones at 860 nm under a bias of −0.1 V and a frequency of 165 Hz (see Supplementary Fig. 11). The EQE values under various illumination intensity is also a critical performance parameter for photodiodes, and the result for the double donor layer device is shown in Supplementary Fig. 12. These critical parameters, including pretty high peak EQE value, responsivity and $D^*$ value, make our device perform much better than previously reported no-gain type narrowband OPDs, and even comparable to commercialized silicon photodetectors (see Supplementary Fig. 13).

**Universality of self-filtering narrowband OPD structure**. The generic device structure proposed in this study is also applicable to other common organic semiconductor materials. The position and full-width-at-half-maximum (FWHM) of the response peaks can both be tuned by simply adjusting the combination of the donor and the acceptor materials in this study. Generally, when adopt donor and acceptor materials with red-shifted absorption onsets, the response peak will simultaneously red-shift, and when the absorption spectra of donor and acceptor materials overlap more with each other, the FWHM will be narrowed. Take the device structure based on P3HT as the SF-HTL (as shown in the Fig. 4a) for an example. The selected donor materials and acceptor materials are summarized in the Fig. 4b. As the normalized EQE spectra shown in the Fig. 4c, keeping the acceptor material Y6 unchanged, when the donor material NT812 was replaced by DT-PDPP2T-TT[52], whose absorption onset is further redshifted than NT812 (Supplementary Fig. 14a), the FWHM of response peak was compressed from 72 to 43 nm, and the peak position was also red shifted from 860 to 910 nm. Furthermore, based on this donor front layer, when the acceptor material was changed from Y6 to IEICO-4F[53], whose absorption onset is further redshifted than Y6 (Supplementary Fig. 14b), the response peak was further red shifted to 940 nm with a FWHM of 66 nm. And as shown in Fig. 4d, both photodetectors exhibit extremely low dark current density, delivering a peak specific detectivity $D^*$ about $10^{13}$ Jones in their respective detection windows (as shown in Fig. 4e, f, corresponding responsivity are shown in Supplementary Fig. 15), demonstrating the good universality of this EDN approach.

## Discussion

In conclusion, we successfully constructed narrowband OPDs by addressing the disadvantages of a large binding energy and the low diffusion length of photogenerated Frenkel excitons in organic semiconductors. By calculating the diffusion length and hence the distribution of the photogenerated excitons in the donor layer, we could intentionally manipulate the dissociation efficiency of excitons generated by various wavelengths of incident light via a hierarchical device structure. The SF-narrowband OPDs were thus achieved via the concept of EDN. This strategy avoids the undesirable sensitivity degradation accompanied with the thick BHJ. Compared with conventional thin junction device under the same operating voltage, as the response in the visible range is completely suppressed, the peak EQE in the narrow NIR detection window still retains more than 89% of the corresponding value. Moreover, the utilizing of Frenkel exciton endows the resulting detector with electrically stable spectral selectivity, thus the device can still maintain narrow response even at −10 V bias, achieving a peak EQE value around 65%. In addition, the multilayer structure improved the charge injection barrier and effectively suppressed dark current, leading to higher detectivity. As consequence, a series of visible-blind NIR narrowband OPDs with FWHM of around 50 nm, peak detectivity over $10^{13}$ Jones were demonstrated. This novel device structure along with its generic design concept may endow organic semiconductors with the ability to create filter-free narrowband OPDs for practical applications, examples include active imaging for face identification in smart phones, for augmented/virtual reality goggles and for full-weather robots. The narrow wavelength selectivity at 860, 910, and 940 nm and the corresponding performance parameters demonstrated in this work provide an "ideal" photodetector for artificial intelligent applications. It should be noted that, different from the response peak which is highly dependent on the absorption spectra of donor and acceptor, the EQE and $D^*$ are affected by the photon–electron conversion efficiency of donor and acceptor combination. Therefore, narrowband OPDs with desired response peak, high EQE and $D^*$ can be expected by selecting existing donor and acceptor materials, or designing new materials (needs to co-work with material scientist).

## Methods

**Materials**. P3HT, DT-PDPP2T-TT, IEICO-4F, Y6, and all reagents were purchased from commercial sources (1-Material and Solarmer Materials Inc.) and used without further purification. The polymer NT812 and the crosslinker s-4FPA were synthesized in-house following established procedures in the refs. [30,37].

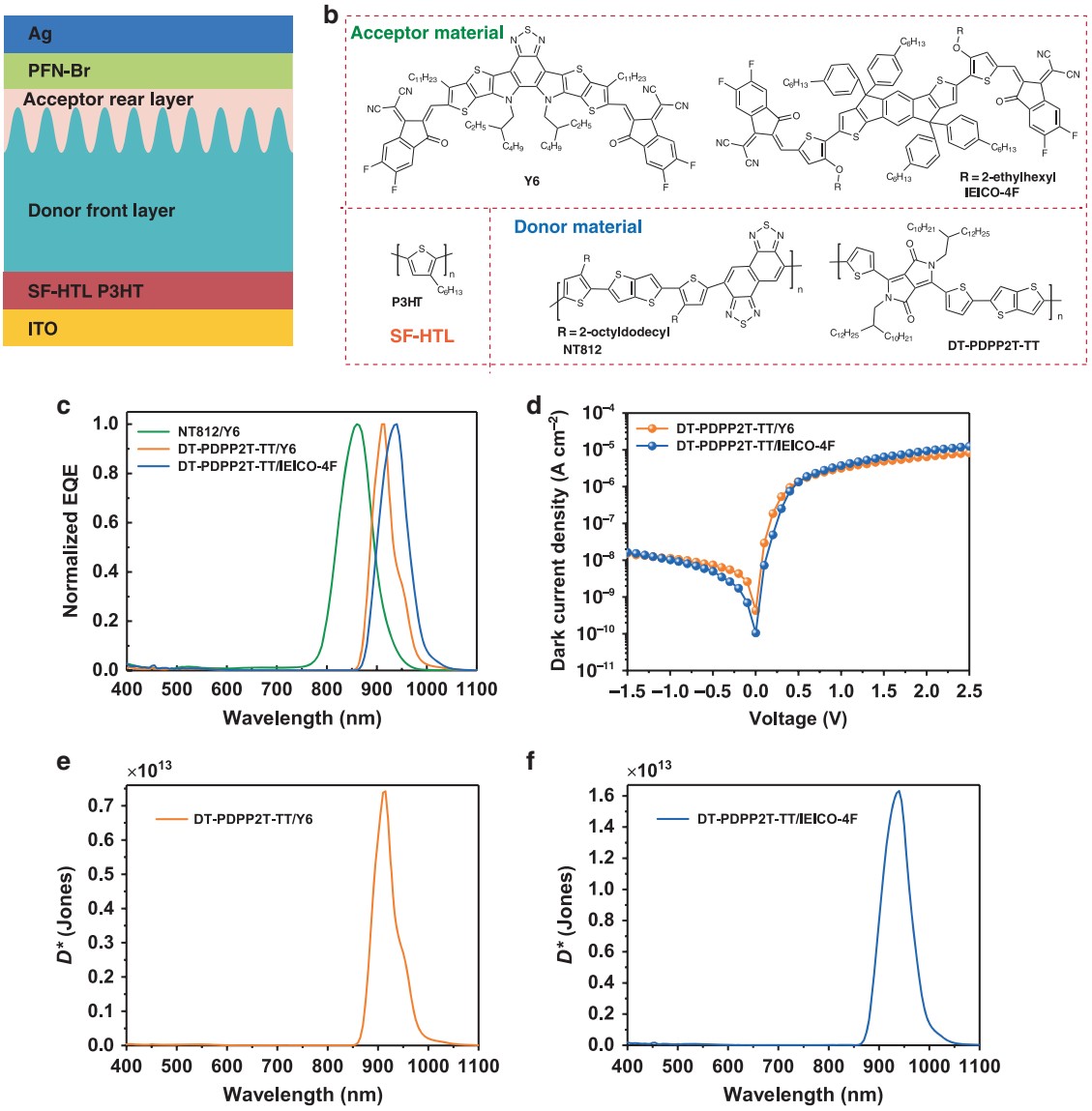

**Fig. 4 Universality of the SF-narrowband OPD. a** Typical device structure of self-filtering (SF) narrowband organic photodetector (OPD) with double donor layers. **b** Chemical structures of the selected donor materials, acceptor materials and SF hole transport layer (SF-HTL) materials. **c** Normalized external quantum efficiency (EQE) spectra of SF-narrowband OPDs fabricated by matching different materials, all devices were measured at −0.1 V bias. **d** Dark current density of SF-narrowband OPDs based on DT-PDPP2T-TT/Y6 and DT-PDPP2T-TT/IEICO-4F. **e** Specific detectivity spectra of OPDs based on DT-PDPP2T-TT/Y6 and **f** DT-PDPP2T-TT/IEICO-4F, respectively.

**Device fabrication**. ITO-coated glass was used as the substrate. Before device fabrication, the substrates were thoroughly cleaned by sequentially sonication with acetone, isopropanol, detergent, de-ionized water, isopropanol, and subsequently dried in a baking oven over night. After that, the substrates were treated by oxygen plasma for 4 min. For single donor layer device, PEDOT:PSS (P VP Al 4083) of 30 nm was first spun onto the substrates and then annealed at 150 °C on a hot plate for 20 min in air to remove the residual water. NT812 mixed with 10 wt% s-4FPA was dissolved in CB solvent, the blend of NT812 and s-4FPA was spun onto the PEDOT:PSS, annealed at 90 °C for 5 min in the glove box with nitrogen atmosphere, photo-exposed to deep-ultraviolet light (DUV, 254 nm wavelength), a crosslinked film of about 650 nm was then obtained. NT812 was dissolved in CB solution, and then spun onto the crosslinked film to obtain a thin film of about 100 nm. Y6 was dissolved in CF solvent, and then spun onto the donor front layer to obtain a complete activelayer with a total thickness of 800 nm. After spin-coating of 8 nm PFN-Br as cathode interface, a 100 nm Ag layer was sequentially deposited by thermal evaporation through a shadow mask in a vacuum chamber at a pressure of $4 \times 10^{-7}$ torr. For double donor layer devices based on NT812/Y6, P3HT mixed with 12 wt% s-4FPA was dissolved in CB solvent, the blend of P3HT and s-4FPA was spun onto the ITO substrates, annealed at 90 °C for 5 min in the glove box with nitrogen atmosphere, photo-exposed to deep-ultraviolet light (DUV, 254 nm

wavelength), and then the film was developed with CF to obtain the SF-HTL with a thickness of 150 nm. The following fabrication steps were the same as in single donor layer devices. For double donor layer devices based on DT-PDPP2T-TT/Y6, DT-PDPP2T-T mixed with 10 wt% s-4FPA was dissolved in CF solvent, the blend of DT-PDPP2T-T and s-4FPA was spun onto the SF-HTL P3HT (200 nm), photo-exposed to deep-ultraviolet light (DUV, 254 nm wavelength), a crosslinked DT-PDPP2T-T film of about 1400 nm was then obtained. DT-PDPP2T-T was dissolved in CF solution, and then spun onto the crosslinked film to obtain a thin film of about 100 nm. Y6 was dissolved in CF solvent, and then spun onto the donor front layer to obtain a complete activelayer with a total thickness of 1800 nm. For double donor layer device based on DT-PDPP2T-TT/IEICO-4F, the fabrication methods of 200 nm SF-HTL P3HT/1400 nm crosslinked DT-PDPP2T-T/150 nm DT-PDPP2T-T were the same as DT-PDPP2T-TT/Y6-based device, IEICO-4F was dissolved in CF, then spun onto the donor front layer to obtain a complete activelayer with a total thickness of 2000 nm. The following fabrication steps were the same as in single donor layer devices. The device active area was 0.0516 cm².

**Device characterization**. The thickness of the thin films was determined by a Dektak 150 surface profiler. The EQE spectrum was measured on a commercial

measurement system DSR100UV-B (Zolix Instruments Co., Ltd.) equipped with DC module. The light intensity at each wavelength was calibrated using a standard single crystal Si photovoltaic cell before the testing. The light frequency was 165 Hz. The light intensity was 2.4 μw cm$^{-2}$ at 860 nm, 2.8 μw cm$^{-2}$ at 910 nm and 3.1 μw cm$^{-2}$ at 940 nm, respectively. The dark current density–voltage characteristics of the devices were recorded on a Keithley 2450 source-meter in an electrically and optically shielded box. The noise spectral density characteristics of the devices were recorded by a semiconductor parameter analyzer (Platform Design Automation, Inc. FS380 Pro). The frequency dependent measurements were carried out using a light-emitting diode (860 nm) modulated by the function generator (Aim-TTi TG120) as the excitation source. Square waves with different frequencies were applied. The photocurrent response of the photodiode was recorded using a digital storage oscilloscope (Tektronix TDS3052B).

**ToF-SIMS depth profiling**. A ToF-SIMS 5–100 (ION-TOF GmbH, Germany) instrument was used to acquire depth profiles from the prepared film. The instrument was equipped with a Bi/Mn liquid metal ion gun (LMIG) and an argon gas cluster ion gun, which were operated in the dual beam mode. For depth profiling, a 2.5 keV Ar-Cluster beam was used to sputter through the film at an area of 300 μm × 300 μm in 5 s intervals. A 25 keV Bi$_3^+$ analysis beam was used to analyze the central area between sputtering pulses over a 100 μm × 100 μm area inside the crater. The negative polarity data were used for sample analysis. The three-dimensional secondary ion images were reconstructed from the depth profile data.

**Time-resolved photoluminescence (TRPL)**. The TRPL measurements were carried out by time-correlated single photon counting (TCSPC) technique: Acton SP2150i spectrometer (Princeton instrument) equipped with a photomultiplier (PMA 182-N-M) were used to collect the photons, an event timer (HydraHarp-400 TCSPC) with 2 ps time resolution was used to measure the fluorescence decays, a 670 nm pulse laser was generated by the 80 MHz femtosecond laser with full width at half maximum about 120 fs, and the decay time fitting procedure was realized by the software FluoFit applying deconvolution with the instrument response function (IRF).

**UV–Vis absorption and transmittance spectra**. UV–Vis absorption spectra were acquired on a Shimadzu UV3600 spectrophotometer. Transmittance spectra were measured on an HP 8453E spectrophotometer.

**Scanning electron microscope**. The scanning electron microscope images were obtained with a Hitachi S-4800 FESEM.

## Data availability
The data that support the findings of this study are available from the corresponding author upon reasonable request.

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

## Acknowledgements

This work was financially supported by the National Key Research and Development Program of China (No. 2019YFA0705900) funded by MOST, the Natural Science Foundation of China (Nos. 21520102006), and Guangdong Major Project of Basic and Applied Basic Research (No. 2019B030302007).

## Author contributions

K.Z. and F.H. conceived the ideas and coordinated the work. B.X. carried out the device fabrication and characterization, performed the measurements of SCLC, UV–Vis absorption and transmittance spectra, performed Monte Carlo simulation and the calculation of exciton concentration distribution, and analyzed the data. Q.Y. conducted the TRPL measurement and analyzed the TRPL data. R.X. synthesized NT812. Z.H. synthesized s-4FPA. B.X., K.Z., G.Y., F.H. and Y.C. contributed to manuscript preparation. All authors commented on the manuscript.

## Competing interests

The authors declare no competing interests.
