## [Peer Review File · Nature Communications]

Reviewers' Comments:

Reviewer #1:

Remarks to the Author:

This manuscript introduces a new concept for making submicron thick narrowband organic photodetectors with a high peak external quantum efficiency. The photodiodes consist of a multilayer architecture, with subsequently spincoated layers on top of each other. A thicker (~800 nm) polymer layer is followed by a thinner small molecule acceptor layer. Low energy photons do reach the acceptor layer and produce a photocurrent at high quantum efficiency (>50%). Most of the excitons generated in the high gap polymer do not reach the polymer:acceptor interface and thus do not contribute to photocurrent. This introduces a filter effect, enabling narrowband detection.

This is an original and useful concept, the experiments are executed carefully and appropriately, and the manuscript is well structured. I recommend publication after authors take into account the following comments:

- P3: "a localized Frenkel exciton with a radius of two or more lattice constants is created ...". What would be the lattice constant in these amorphous polymers? Is it even possible to define a lattice constant?
- Legend of figure 2: Frenkle exciton  Frenkel exciton
- Authors refer to n-type and p-type layers. It would be much better to refer to acceptor and donor materials, or electron and hole transporting materials. An n-type (or p-type) material implies deliberate doping to introduce mobile electrons (or holes). As there is no deliberate doping of these materials done, it is unknown if they are truly n-type (or p-type) doped.
- It would be useful to see the specific detectivity vs wavelength plots on a logarithmic scale. On a log scale, its more clear how many orders of magnitude the response of the filtered out wavelengths is reduced. For some applications these unwanted wavelengths should be rejected by a factor of 100 or more (EQE in unwanted region is < 0.01*EQE in peak region).
- The specific detectivity is determined at -0.1 V. In this region, there is still a substantial slope of the dark current vs voltage curve, implying that the thermal noise $\sim\sqrt{\text{slope of IV curve}}$ is non-negligible. Authors should include thermal noise in their calculation of the noise current and detectivity. The measured noise current might be closer to the calculated noise current if thermal noise is included.

Reviewer #2:

Remarks to the Author:

The authors reported a sort of self-filtering near-infrared narrowband organic photodetector with thick larger bandgap p-type layers followed by a lower bandgap n-type layer. The FWHM of the response spectrum is about 50 nm, the peak EQE is about 65% and the peak detectivity is over 10^{13} Jones. The work is interesting. However, more data and explanations of the working mechanism should be added to show the innovation of the device.

1. Several papers also have reported near-infrared narrowband OPDs with thick active layers to absorb visible light and get narrowband near-infrared response. (Jinsong Huang, et al. Nano Lett. 2017, 17, 1995–2002; Fang Ying, et al. Adv. Optical Mater. 2018, 6, 1800001) Their FWHMs are even smaller than 50 nm and the EQEs are larger than 65%. The innovation of the device in this manuscript is not clearly described by the authors.

2. In the manuscript, the near-infrared sensitive peak wavelength is different with different p-type material and different n-type material. However, how the materials influence the properties of the OPDs is not explained.

3. I-V (or J-V) curves in dark and under illumination and the curves of EQE versus the light power are helpful to measure the performance of devices. If possible, the curves should be added in the revised manuscript.

Reviewer #3:

Remarks to the Author:

The paper proposed methods and experimental results to construct self-filtering narrowband organic photodetectors. The hierarchical device structure helps to manipulate the dissociation efficiency of excitons and the exciton dissociation narrowing (EDN) technique is able to retain device high sensitivity at desirable wavelengths. The very high peak EQE in the narrow NIR detection window of FWHM about 50nm and peak detectivity over $E+13$ Jones were achieved. The authors further proposed to apply double p-type layers to enhance suppression of off-band incident light. They also replaced p-type and n-type materials to control the device response peak.

In general, the paper advanced the state-of-the-art in organic NIR photodetectors. The methods, experimental procedure and materials appear feasible. Some suggestions are as follows.

1. Authors mentioned suppression of the response outside the detection window (L69). Since an optical filter usually quantifies the out-of-band suppression ratio, could you give a similar figure of merit to characterize the "self-filtering" property (including in-band transmission and out-of-band rejection)?
2. Authors gave ways to control peak wavelength and FWHM only by changing materials. But it is not really tunable as you did not show how to control them to be in specific wavelength and FWHM. If you change the materials, you in fact change the device. How to make the device tunable? Please revise.
3. When authors discussed tunable response peak, how does it affect the EQE, and D? Do these be sacrificed to achieve your new goals? Could you elaborate this more?

Minor grammar:

L39, respond

L72, performances that even comparable

List of point-by-point response to reviewers' comments

Reviewer #1 (Remarks to the Author):

This manuscript introduces a new concept for making submicron thick narrowband organic photodetectors with a high peak external quantum efficiency. The photodiodes consist of a multilayer architecture, with subsequently spincoated layers on top of each other. A thicker (~800 nm) polymer layer is followed by a thinner small molecule acceptor layer. Low energy photons do reach the acceptor layer and produce a photocurrent at high quantum efficiency (>50%). Most of the excitons generated in the high gap polymer do not reach the polymer:acceptor interface and thus do not contribute to photocurrent. This introduces a filter effect, enabling narrowband detection.

This is an original and useful concept, the experiments are executed carefully and appropriately, and the manuscript is well structured. I recommend publication after authors take into account the following comments:

Reply: We thank the reviewer for his/her positive comments and careful attention to improve the clarity and content of the manuscript. The responses to his/her questions are listed below.

1. P3: “a localized Frenkel exciton with a radius of two or more lattice constants is created ...”. What would be the lattice constant in these amorphous polymers? Is it even possible to define a lattice constant?

Reply: Thank the reviewer for his/her reminding. We agree with the reviewer that it is impossible to define a lattice constant for amorphous polymers. Only for some highly crystalline polymeric semiconductors, lattice constant definition can be used in theoretical research or highly crystalline local area (E. R. Bittner, C. Silva, Noise-induced quantum coherence drives photo-carrier generation dynamics at polymeric semiconductor heterojunctions, Nature Commun., 2014, 5, 3119; M. Pope, C. E. Swenberg, Electronic processes in organic crystals and polymers, Oxford University Press, New York 1982). Since these polymers used in our study have a lot of amorphous region, as the reviewer suggested, we have deleted the description of

lattice constant in our revised manuscript to improve the clarity of this work.

“...a localized Frenkel exciton is created...”

2. Legend of figure 2: Frenkle exciton  Frenkel exciton

Reply: Thank the reviewer for pointing out this written mistake. In our revised manuscript, we have revised Fig. 2c.

3. Authors refer to n-type and p-type layers. It would be much better to refer to acceptor and donor materials, or electron and hole transporting materials. An n-type (or p-type) material implies deliberate doping to introduce mobile electrons (or holes). As there is no deliberate doping of these materials done, it is unknown if they are truly n-type (or p-type) doped.

Reply: Thank the reviewer for this suggestion. As the reviewer said, the organic

semiconductor materials used in this study do not have any deliberate doping treatment, so it is not suitable to call them n-type and p-type materials. Therefore, we take the reviewer's advice and revise the relevant statements in the full text.

In our revised manuscript, we have changed “p-type” to “donor” and “n-type” to “acceptor”.

4. It would be useful to see the specific detectivity vs wavelength plots on a logarithmic scale. On a log scale, its more clear how many orders of magnitude the response of the filtered out wavelengths is reduced. For some applications these unwanted wavelengths should be rejected by a factor of 100 or more (EQE in unwanted region is $< 0.01 \cdot \text{EQE}$ in peak region).

Reply: Thank the reviewer for the suggestion. The specific detectivity vs wavelength plots on a logarithmic scale is presented in the following figure. As expected, the specific detectivity at the detection peak is two orders of magnitude higher than that outside the detection window.

In our revised manuscript, we have added specific detectivity vs wavelength plots on a logarithmic scale in Supplementary Fig. 8, and we have added the following discussion in the revised manuscript.

“The specific detectivity curve on a logarithmic scale in Supplementary Fig. 8 shows that the specific detectivity at the detection peak is two orders of magnitude higher than that outside the detection window.”

Supplementary Figure 8 | Specific detectivity spectra obtained from J_d on a logarithmic scale.

5. The specific detectivity is determined at -0.1 V. In this region, there is still a substantial slope of the dark current vs voltage curve, implying that the thermal noise $\sim \sqrt{\text{slope of IV curve}}$ is non-negligible. Authors should include thermal noise in their calculation of the noise current and detectivity. The measured noise current might be closer to the calculated noise current if thermal noise is included.

Reply: Thank the reviewer for the suggestion. We have added the term of thermal noise (i_{th}) into the dark current noise formula (i_d) and obtained a calculated noise (i_{cal}) and a calculated detectivity (D^). As the reviewer suggested, the following calculation has been added in supplementary information as Supplementary Note 3.*

The magnitude of thermal noise^{s1} (i_{th}) is expressed as Equation (3)

$$i_{th} = \sqrt{\frac{4kTB}{R_L}} \quad (3)$$

where k is the Boltzmann constant, T is temperature in Kelvin, which is 293 K under the experimental condition, B is the noise measurement bandwidth, which is set as 165 Hz, and R_L is the shunt resistance in the device, which is 480 M Ω obtained from the I-V slope.

The magnitude of the dark current noise (i_d) is expressed as Equation (4)

$$i_d = \sqrt{2qI_dB} \quad (4)$$

where q is the electron charge, I_d is the dark current, which is 2.08×10^{-10} A at -0.1V bias.

So the calculated noise current can be expressed as Equation (5)

$$i_{cal} = \sqrt{i_d^2 + i_{th}^2} = 1.29 \times 10^{-13} \text{ A} \quad (5)$$

And the calculated detectivity can be expressed as Equation (6)

$$D^* = \frac{R\sqrt{AB}}{i_{cal}} \quad (6)$$

Where R is the responsivity and A is the device area. The calculated detectivity demonstrates a peak value of 9.5×10^{12} Jones at 860 nm, which is slightly lower than

that obtained without the thermal noise of 1.2×10^{13} Jones.

s1. Yang, D. & Ma, D. Development of organic semiconductor photodetectors: from mechanism to applications. Adv. Optical Mater. 7, 1800522 (2019).

The following description has been added into the revised manuscript.

“When the thermal noise is taken into consideration, the calculated D^ slightly decreases to 9.5×10^{12} Jones at 860 nm, see the Supplementary Note 3”*

Reviewer #2 (Remarks to the Author):

The authors reported a sort of self-filtering near-infrared narrowband organic photodetector with thick larger bandgap p-type layers followed by a lower bandgap n-type layer. The FWHM of the response spectrum is about 50 nm, the peak EQE is about 65% and the peak detectivity is over 10^{13} Jones. The work is interesting. However, more data and explanations of the working mechanism should be added to show the innovation of the device.

Reply: We thank the reviewer for his/her positive comments and careful attention to improve the clarity and content of the manuscript. The responses to his/her questions are listed below.

1. Several papers also have reported near-infrared narrowband OPDs with thick active layers to absorb visible light and get narrowband near-infrared response. (Jinsong Huang, et al. Nano Lett. 2017, 17, 1995–2002; Fang Ying, et al. Adv. Optical Mater. 2018, 6, 1800001) Their FWHMs are even smaller than 50 nm and the EQEs are larger than 65%. The innovation of the device in this manuscript is not clearly described by the authors.

Reply: Thank the reviewer for this question. In this work, we propose a simple strategy to produce narrowband OPDs by manipulating exciton dissociation (named as “exciton dissociation narrowing” [EDN]), with a hierarchical device structure

where larger bandgap donor layers followed by a lower bandgap acceptor layer. During the operation, excitons generated by high-energy photons in front donor layers fail to separate into free charges due to the absence of D/A interfaces, thus dissipated. Only low-energy photons with a long penetration depth can reach the D/A interfaces and produce free charges for collection. This novel methodology can efficiently suppress the response outside the detection window while retaining high sensitivity in the detection region. Some other strategies based on thick active-layer (typically 2 ~ 5 μm) with bulk heterojunction (BHJ) structure have also been proposed to construct narrowband photodetectors, including no-gain type and gain type. The no-gain type narrowband detectors enabled by CCN concept (e.g. the pioneer work made by Armin et. al., Nature Commun. 2015, 6, 6343) modulate the EQE spectrum by manipulating the collection efficiency of free charges, instead of excitons, at the corresponding electrodes. Higher applied voltages can enhance the charge collection efficiency at corresponding electrodes, thereby destroying the narrowband characteristics of CCN devices. While the gain type detectors (e.g. some outstanding works made by Prof. Zhang et. al., Nano Lett. 2017, 17, 1995–2002; Adv. Optical Mater. 2018, 6, 1800001. Cited as ref. 28 and 29 in the revised manuscript) achieve photomultiplication via charge tunneling injection from external circuit under large applied voltage (tens of volts), resulting in a lower detectivity. Our device is different from previously reported works in both device structure and working mechanism, which lead to the device with high detectivity and responsivity, lower film thickness (~ 1 μm), and nearly constant spectral selectivity even under high voltages, etc.

According to the reviewer's suggestion, in the introduction part of our revised manuscript, we have added some more discussions to better illustrate the innovation of this study.

“Some other strategies based on thick active layer with bulk heterojunction (BHJ) structure have also been proposed to construct narrowband photodetectors, including no-gain type¹¹ and gain type^{28, 29}. For example, the no-gain type narrowband detectors enabled by CCN concept modulate the external quantum efficiency (EQE)

spectrum by manipulating the collection efficiency of free charges at the corresponding electrodes (photogenerated excitons firstly dissociate into free charges). The gain type detectors achieve photomultiplication via charge tunneling injection from external circuit under large applied voltage. In comparison, the novel device structure and its generic methodology proposed in this study is based on hierarchical active-layer structure and novel working mechanism of exciton dissociation narrowing.”

28. Miao, J., Zhang, F., Du, M., Wang, W. & Fang, Y. Photomultiplication type organic photodetectors with broadband and narrowband response ability. Adv. Optical Mater. 6, 1800001 (2018).

29. Wang, W. et al. Highly narrowband photomultiplication type organic photodetectors. Nano Lett. 17, 1995–2002 (2017).

2. In the manuscript, the near-infrared sensitive peak wavelength is different with different p-type material and different n-type material. However, how the materials influence the properties of the OPDs is not explained.

Reply: Thank the reviewer for his/her questions. In our revised manuscript, we have made some revisions to explain how the materials influence the properties of the OPDs to improve the clarity of this manuscript.

“The position and full-width-at-half-maximum (FWHM) of the response peaks can both be tuned by simply adjusting the combination of the donor and the acceptor materials in this study. Generally, when adopt donor and acceptor materials with red-shifted absorption onsets, the response peak will simultaneously red-shift, and when the absorption spectra of donor and acceptor materials overlap more with each other, the FWHM will be narrowed. Take the device structure based on P3HT as the SF-HTL (as shown in the Fig. 4a) for an example. The selected donor materials and acceptor materials are summarized in the Fig. 4b. As the normalized EQE spectra shown in the Fig. 4c, keeping the acceptor material Y6 unchanged, when the donor

material NT812 was replaced by DT-PDPP2T-TT⁵¹, whose absorption onset is further redshifted than NT812 (Supplementary Fig. 13a), the FWHM of response peak was compressed from 72 nm to 43 nm, and the peak position was also red shifted from 860 nm to 910 nm....”

Supplementary Figure 13| Universality of self-filtering narrowband OPDs. Normalized UV-Vis absorption spectra of (a) donor materials and (b) acceptor materials.

3. I-V (or J-V) curves in dark and under illumination and the curves of EQE versus the light power are helpful to measure the performance of devices. If possible, the curves should be added in the revised manuscript.

Reply: Thank the reviewer for the suggestion. In our revised manuscript, we have added the J-V curve under illumination in Fig.3e. Our device based on hierarchical active-layer structure shows great potential in suppressing the dark current.

Figure 3 (e) *J-V* curves in dark and under illumination.

We also added curve of EQE versus the light intensity in Supplementary Fig. 11. The device maintains high EQE with light powers varied over 6 orders.

Supplementary Figure 11 | The curve of EQE versus the light intensity at 850 nm under $-0.1V$ bias.

Reviewer #3 (Remarks to the Author):

The paper proposed methods and experimental results to construct self-filtering narrowband organic photodetectors. The hierarchical device structure helps to manipulate the dissociation efficiency of excitons and the exciton dissociation

narrowing (EDN) technique is able to retain device high sensitivity at desirable wavelengths. The very high peak EQE in the narrow NIR detection window of FWHM about 50nm and peak detectivity over E+13 Jones were achieved. The authors further proposed to apply double p-type layers to enhance suppression of off-band incident light. They also replaced p-type and n-type materials to control the device response peak.

In general, the paper advanced the state-of-the-art in organic NIR photodetectors. The methods, experimental procedure and materials appear feasible. Some suggestions are as follows.

Reply: We thank the reviewer for his/her positive comments and careful attention to improve the clarity and content of the manuscript. The responses to his/her questions are listed below.

1. Authors mentioned suppression of the response outside the detection window (L69). Since an optical filter usually quantifies the out-of-band suppression ratio, could you give a similar figure of merit to characterize the “self-filtering” property (including in-band transmission and out-of-band rejection)?

Reply: Thank the reviewer for the question. As the reviewer suggested, we have made some calculations and revisions to further clarify the “self-filtering” performance of the device.

“Here, we define two parameters, out-of-band suppression factor $S = \frac{\text{nor. } R_{\text{ref}}(\lambda)}{\text{nor. } R_{\text{SF}}(\lambda)}$ and in-band transmission factor $T = \frac{R_{\text{ref}}(\lambda_0)}{R_{\text{SF}}(\lambda_0)}$, to characterize the self-filtering property of the narrowband photodetector, where $\text{nor. } R_{\text{ref}}(\lambda)$ and $\text{nor. } R_{\text{SF}}(\lambda)$ are the normalized responsivity of the reference device and SF-narrowband OPD, respectively. $R_{\text{ref}}(\lambda_0)$ and $R_{\text{SF}}(\lambda_0)$ are the responsivity of the reference device and SF-narrowband OPD at the narrow respond peak, respectively. Results are shown in Supplementary Fig. 6. The maximum out-of-band suppression factor at visible region is over 170, while the in-band transmission factor at 860 nm is 89.2%.”

Supplementary Figure 6 | Characterization of self-filtering property. Responsivity of the reference device (green solid line) and the SF-narrowband OPD with double donor layers (blue solid line) under -0.1V bias; the curve of S (red dashed line) and the value of T at 860 nm.

2. Authors gave ways to control peak wavelength and FWHM only by changing materials. But it is not really tunable as you did not show how to control them to be in specific wavelength and FWHM. If you change the materials, you in fact change the device. How to make the device tunable? Please revise.

Reply: Thank the reviewer for his/her reminding. In our previous manuscript, when we talk about “tunable”, what we want to express is that the device structure and work mechanism proposed in this study is also applicable in other common organic semiconductor materials. Since the absorption spectrum of organic semiconductors can be easily tuned by adjusting their molecular structures, the peak wavelength and FWHM of the device can be tuned by simply adjusting the combination of the donor and the acceptor materials. So we call it tunable. We are sorry for this misleading. As the reviewer suggested, in our revised manuscript, in order to avoid confusion and misleading, we have revised the discussion on “tunable response peak and FWHM” in our revised manuscript into the “universality of self-filtering narrowband OPD

structure”.

3. When authors discussed tunable response peak, how does it affect the EQE, and D? Do these be sacrificed to achieve your new goals? Could you elaborate this more?

Reply: Thank the reviewer for this question. The response peak of self-filtering narrowband OPDs is highly dependent on the absorption spectra of donor and acceptor as discussed in the manuscript. In terms of the EQE and D, which are affected by the photon-electron conversion efficiency of donor and acceptor combination. In our works, we tried some other combinations such as DT-PDPP2T-TT/Y6 and DT-PDPP2T-TT/IEICO-4F. From these results, the response peaks and FWHMs of these two devices are apparently changed in comparison with NT812/Y6 while the detectivities are comparable ($\sim 10^{13}$ Jones) with that of NT812/Y6 device. We believe that self-filtering narrowband OPDs with desired response peak, high EQE and D* can be obtained by selecting existing donor and acceptor materials, or designing new materials (needs to co-work with material scientist).*

We have made some revisions in order to further improve the clarity of this manuscript.

“It should be noted that, different from the response peak which is highly dependent on the absorption spectra of donor and acceptor, the EQE and D are affected by the photon-electron conversion efficiency of donor and acceptor combination. Therefore, narrowband OPDs with desired response peak, high EQE and D* can be expected by selecting existing donor and acceptor materials, or designing new materials (needs to co-work with material scientist).”*

4. Minor grammar:

L39, respond

L72, performances that even comparable

Reply: Thank the reviewer for pointing out these grammar mistakes.

We have revised “respond to” into “response of”.

And we have revised “...with outstanding performances that even comparable to

silicon photodetectors” into “...with outstanding performances”.

Reviewers' Comments:

Reviewer #1:

Remarks to the Author:

The authors have taken into account all my comments and have revised the manuscript accordingly. I recommend publication.

Reviewer #2:

Remarks to the Author:

The authors revised the manuscript carefully according to the reviewers comments. I would like to recommend the revised manuscript be accepted by Nature Communication journal.

Reviewer #3:

Remarks to the Author:

Authors have addressed my comments, particularly provided out-of-band suppression factor and results which show satisfactorily one to two orders of magnitude as compared with in-band transmission, and clarified tunability, EQE, and D. I am satisfied with the revision. Meanwhile, I looked through authors' replies to other two reviewers' comments and checked corresponding revisions. They looked good to me.

I suggest to accept this version.

REVIEWERS' COMMENTS:

Reviewer #1 (Remarks to the Author):

The authors have taken into account all my comments and have revised the manuscript accordingly. I recommend publication.

Response: We thank the reviewer for his/her positive comments for this work.

Reviewer #2 (Remarks to the Author):

The authors revised the manuscript carefully according to the reviewers comments. I would like to recommend the revised manuscript be accepted by Nature Communication journal.

Response: We thank the reviewer for his/her positive comments for this work.

Reviewer #3 (Remarks to the Author):

Authors have addressed my comments, particularly provided out-of-band suppression factor and results which show satisfactorily one to two orders of magnitude as compared with in-band transmission, and clarified tunability, EQE, and D. I am satisfied with the revision. Meanwhile, I looked through authors' replies to other two reviewers' comments and checked corresponding revisions. They looked good to me.

I suggest to accept this version.

Response: We thank the reviewer for his/her positive comments for this work.